# Gaps and opportunities for data systems and economics to support priority setting for climate-sensitive infectious diseases in sub-Saharan Africa: A rapid scoping review

Ellie A. Delight[1]*, Ariel A. Brunn[2], Francis Ruiz[3], Jessica Gerard[1,4], Jane Falconer[5], Yang Liu[6], Bubacarr Bah[7], Bernard Bett[8], Benjamin Uzochukwu[9], Oladeji K. Oloko[10], Esther Njuguna[11], Kris A. Murray[10]

1 Department of Disease Control, London School of Hygiene and Tropical Medicine, London, United Kingdom, 2 Department of Population Health, London School of Hygiene and Tropical Medicine, London, United Kingdom, 3 Department of Global Health and Development, London School of Hygiene and Tropical Medicine, London, United Kingdom, 4 Centre on Climate Change and Planetary Health, London School of Hygiene and Tropical Medicine, London, United Kingdom, 5 Library, Archive & Open Research Services, London School of Hygiene and Tropical Medicine, London, United Kingdom, 6 Department of Infectious Disease Epidemiology and Dynamics, London School of Hygiene and Tropical Medicine, London, United Kingdom, 7 Data Science Cluster, Medical Research Council Unit The Gambia at the London School of Hygiene and Tropical Medicine, Banjul, The Gambia, 8 Education and Outreach Centre for Africa, International Livestock Research Institute, Nairobi, Kenya, 9 Department of Community Medicine, University of Nigeria, Nsukka, Nigeria, 10 Centre on Climate Change and Planetary Health, MRC Unit The Gambia at London School of Hygiene and Tropical Medicine, Banjul, The Gambia, 11 African Social and Gender Insights Group, Nairobi, Kenya

* ellie.delight@lshtm.ac.uk

## Abstract

Climate change alters risks associated with climate-sensitive infectious diseases (CSIDs) with pandemic potential. This poses additional threats to already vulnerable populations, further amplified by social factors such as gender inequalities. Currently, critical evidence gaps, along with inadequate institutional and governance mechanisms, hinder African states' ability to prevent, detect and respond to CSIDs. Effective responses require transparent and evidence-based decision-making processes, supported by fit-for-purpose data systems and robust economic analyses. The aim of this study was to explore the role of data systems and economics in priority setting for CSID pandemic preparedness in sub-Saharan Africa. We conducted a rapid scoping review following PRISMA-ScR guidelines. A literature search was performed across six bibliographic databases in November 2023. A list of 14 target CSIDs was produced, informed by the World Health Organization's Public Health Emergencies of International Concern and R&D Blueprint Pathogen lists, and a database of CSIDs. Studies were included if published between 2010 and 2023, were relevant to sub-Saharan Africa, pandemic preparedness, and a target CSID, and applied or assessed economic evaluations or data systems. Extracted data were synthesised using bibliometric analysis, topic categorisation, and a narrative synthesis including

**Data availability statement:** The data is available at the Open Science Framework at https://osf.io/fn8rw/.

**Funding:** This work was supported by the International Development Research Centre (Grant Recipient: Kris A. Murray; Grant Number: PO5001440). The study was conducted independently from the International Development Research Centre.

**Competing interests:** All authors declare that they have no actual or potential competing interests that could have influenced the work reported herein.

the application of a gender lens. We identified 68 relevant studies. Data system studies (n = 50) showed broad coverage across target CSIDs and the WHO AFRO region but also a high degree of heterogeneity, which may indicate a lack of clearly defined standards or research priorities. Economic studies (n = 18) primarily focused on COVID-19 or Ebola and mostly originated from South Africa. Both data system and economic studies identified limited interoperability across sectors and showed a notable absence of gendered considerations. These gaps present important opportunities to strengthen priority setting during pandemics and may contribute to improved and equitable health outcomes.

## Introduction

Rising greenhouse gas emissions impact the frequency and intensity of climate hazards (e.g., droughts, floods, and heatwaves), which can aggravate the transmission of climate-sensitive infectious diseases (CSIDs) [1]. Additionally, changing climate patterns alter ecological niches where humans, animals and vectors are able to survive [2]. In conjunction with changing land-use patterns, these altered ecological settings and climate hazards can impact pathogen exchange among previously isolated vectors, wildlife, livestock, and human populations [3,4]. This can expose immunologically naïve populations to CSIDs, leading to increased risk of CSID pandemics. Notably, all eight of the viral diseases that have been declared as Public Health Emergencies of International Concern (PHEIC) by the World Health Organization (WHO) [5] are due to pathogens sensitive to climate variables [1]. Additionally, eight of the nine pathogens prioritised by the WHO Research & Development (R&D) Blueprint list are classed as climate-sensitive [1,6], with the ninth being a hypothetical pathogen "Disease X". The R&D list helps to prioritise resources towards pathogens that pose significant public health risks due to their epidemic potential or insufficient control measures. This review covered the years 2010 – 2023, encompassing the period of all PHEIC declarations made by the WHO up to 2023, including Influenza and COVID-19.

Vulnerability to CSIDs is exacerbated by marginalisation, with gender inequalities playing an important role. The gendered risks of CSIDs were evident during the COVID-19 pandemic, where the case fatality rate was higher among men [7], but women faced greater socioeconomic impacts and increased risk of domestic violence as a result of lockdown restrictions [8,9]. Given that economic evaluations and data systems often inform policy responses, it is important to assess whether gendered vulnerabilities are adequately considered within these frameworks. If gender is overlooked in these domains, critical disparities may go unrecognised, limiting the effectiveness and equity of interventions. Investigating gendered impacts of pandemics can enable a more contextualised and thorough understanding of the aetiology of CSIDs and facilitate equitable responses tailored to men, women, boys, girls, and other gender identities [10].

A growing body of evidence demonstrates the disproportionate impact of climate change across countries in sub-Saharan Africa, despite minimal contributions to

green house gas emissions [11–13]. Some limited studies have evaluated the risk of CSIDs in sub-Saharan Africa; for instance, in low-income, rural communities in the Sahel region of West Africa, daily high temperatures (above 41.1 °C compared to 36.4°C median) and low rainfall (below 10mm compared to 14mm median) were associated with increased deaths from CSIDs [13]. Additionally, climate hazards such as flooding can prevent access to healthcare services, which can cause diagnostic and treatment delays [14], impeding early detection of outbreaks and raising the risk of uncontrolled disease spread and pandemic emergence.

The ability to prepare for the outsized risks of CSID pandemics in sub-Saharan Africa is limited by insufficient institutional collaborations and integrated governance mechanisms. This hinders data-driven decision-making and appropriate and judicious allocation of scarce resources. Successful policy response requires capacity on both sides of the research-policy nexus, including skilled researchers generating robust data and relevant economic evaluation, and presence of formalised systems or bodies to build confidence and public support for the deployment of public funds during a public health crisis. Interoperable data systems that provide near real-time information across human, animal, and environmental domains inform decision-making for the detection and prevention of CSIDs. Additionally, data systems can help to inform decision-making across later stages of pandemic preparedness; for example by helping to tailor control strategies to regional climate differences that would otherwise hamper response strategies [15].

Priority setting is a process that can help to manage complex decision-making and may be especially beneficial during pandemic response, particularly in low-resource settings where policy makers must balance emergency needs with regular health programs. Recent outbreaks, such as the 2014 Ebola outbreak in West Africa [16] and more recently with COVID-19 globally, illustrate the long-lasting economic and health consequences of pandemics. These impacts are not experienced equally across genders, with women and other marginalised groups facing disproportionate economic losses, particularly seen during the COVID-19 pandemic [17]. Understanding the cost-effectiveness of interventions and response can aid decision-makers in allocating scarce resources during pandemic response. This can incorporate economic evaluation (such as cost-effectiveness analyses (CEA), cost-utility analyses (CUA), or cost-benefit analyses (CBA)), which are ideally institutionalised and include approaches such as Health Technology Assessments (HTA) that consider a body of evidence and multi-stakeholder perspectives.

Effective priority setting with respect to pandemic preparedness and CSIDs, or indeed in any situation where a resource allocation decision needs to be made, should be informed by evidence of both likely benefits and cost-effectiveness [18]. In turn, such assessments of value depend on credible sources of information, even though assumptions and the need to make qualitative judgements are unavoidable. Priority setting is therefore facilitated by having fit-for-purpose (and cost-effective) data systems that capture contextualised information on epidemiology, vulnerable groups, resource use, gender disparities, and other relevant socioeconomic characteristics. As such, data systems are key to informing CSID control strategies, and supporting the parameterisation of economic evaluations that help to consider the benefits of available strategies against the resources required to implement them.

To identify existing knowledge and gaps relevant to data system and economic studies for priority setting of CSIDs with pandemic potential, we adapted a conceptual framework based on four stages of pandemic preparedness identified by the WHO [19]: Prevention (Stage 1): pre-epidemic preparedness; Detection (Stage 2): identify, investigate, evaluate risk; Response (Stage 3): outbreak response & containment; Evaluation (Stage 4): Post-epidemic evaluation (S1 Fig). We used this framework to identify the degree of research intensity in each stage of pandemic preparedness as it relates to our study themes.

## Objectives

i. Map key themes that describe how economic analyses and data systems support priority setting during pandemic preparedness of CSIDs in sub-Saharan Africa, using bibliometrics analysis, topic mapping and narrative synthesis.

ii.  Identify evidence gaps and future research needs to inform priority setting through the use of economic analyses and data systems.

iii. Describe subnational, national, and regional structures in place to support joined-up data sharing for pandemic preparedness of CSIDs.

iv.  Understand the extent to which gender considerations are incorporated into economic analyses and data system studies related to pandemic preparedness of CSIDs.

## Methods

### Rapid scoping review

The literature review was conducted as a rapid scoping review between November 2023 – April 2024. Scoping reviews provide a structured approach to mapping existing knowledge, highlighting common themes, and identifying evidence gaps. Rapid reviews streamline these methods, facilitating quicker knowledge translation, especially when urgent policy or strategy decisions are required [20].

### Protocol and registration

This rapid scoping review was informed by the Preferred Reporting Items for Systematic Reviews and Meta-Analyses for Scoping Reviews (PRISMA-ScR). The protocol for this study is registered at the Open Science Framework (OSF) repository [21] and is available on MedRxiv [22].

### Eligibility criteria

The eligibility criteria are outlined in Table 1.

### Information sources and search

The search strategy was constructed by a library information professional with a focus on four search concepts, namely:

• Pandemic preparedness AND

• Climate change AND

• Economic evaluation (including HTAs and priority setting) OR

• Data systems

  The search was conducted in November 2023 across six bibliographic databases: OvidSP Medline, OvidSP Embase, OvidSP Global Health, EBSCOhost Africa-Wide Information, OvidSP Econlit and Clarivate Analytics Web of Science Core Content. Search terms were first tested in one database prior to implementation in the other five. Additional sources of literature from work previously conducted by co-authors were also included. The complete search strategies for all sources is published at the London School of Hygiene & Tropical Medicine Data Compass [24].

### Selection of sources of evidence

Articles were deduplicated and imported into Covidence software [25]. Screening was conducted by single independent reviewers through a three-stage process: title, abstract, and full text screening. At each stage, articles were screened using the "Most relevant" option in Covidence, which employs a machine learning algorithm to predict study relevance based on screening of at least 25 studies [26].

**Table 1. Scoping review eligibility criteria.**

| Criteria | Inclusion | Exclusion |
|---|---|---|
| **Publication Period** | 2010 - 2023 | Articles outside this period. |
| **Location** | Articles relevant to and published in or about countries or regions in sub-Saharan Africa, defined as any member states of the WHO African Region (AFRO) [23]. | Articles relevant to or published outside of sub-Saharan Africa, articles not geographically located, or articles on globally aggregated data. |
| **Language** | Articles in English. | Non-English language articles were excluded. |
| **Study Type** | Original research, literature reviews, reports, policy briefs, opinion pieces and editorials. | News articles and conference abstracts. |
| **Pandemic Preparedness** | Articles that assessed preparedness that can be contextualised within one of the four stages of adapted conceptual framework (S1 Fig), including Prevention (Stage 1); Detection (Stage 2); Response (Stage 3); and Evaluation (Stage 4). | NA |
| **Climate-sensitive infectious diseases** | Articles referring to CSIDs with pandemic potential. Infectious diseases with pandemic potential were derived from WHO R&D Blueprint Disease List [6] and pathogens declared in PHEIC disease outbreaks [5]. These were compared with a database of CSIDs to confirm their climate sensitivity [1]. Climate sensitivity is based on published associations with climate change variables, including weather, hydrometeorological hazards, and land-use changes over short or long (decadal) timescales. Zoonotic CSIDs that have established a human-to-human transmission cycle are also included if the original pathogen transmission was considered climate-sensitive. CSIDs included in this review are:• Influenza A<br>• Ebola Virus Disease<br>• Marburg Virus Disease<br>• Zika Virus Disease<br>• Severe Acute Respiratory Syndrome (SARS)<br>• Middle East Respiratory Syndrome (MERS)<br>• Coronavirus Disease 2019 (COVID-19)<br>• Crimean-Congo Haemorrhagic Fever<br>• Lassa Fever<br>• Nipah Virus Disease<br>• Henipavirus Disease<br>• Rift Valley Fever<br>• Mpox<br>• Disease X*<br>*Disease X, a hypothetical R&D blueprint pathogen, was included as it could be a zoonotic climate sensitive infectious disease. | Articles that consider climate change-related hazards such as extreme weather as a barrier to healthcare facility access were excluded. Articles that refer to climate in a cultural sense, i.e., political climate, were excluded. Articles about allergens, fungal diseases and antimicrobial resistance are excluded in this review, although it is acknowledged that these disease processes are increasingly being linked to climate change. |
| **Health Economic Theme** | Articles that described the use of HTAs, CUAs, CEAs, priority setting, and costing studies in the context of pandemic preparedness and climate change. | Articles that did not provide analysis or discussion in the context of pandemic preparedness or climate change. |
| **Data Systems Theme** | Articles that used data from established data systems, or that describe the creation of new data systems, or that outline the interoperable use of multi-sectoral data systems. | Articles that mentioned data or information systems not relevant to a health or health economics context. |
| **Outcomes** | Descriptions of frameworks, data system or health information system structures, case study outcomes, trial outcomes, best practices, evidence syntheses including meta-analyses results and cost-effectiveness findings. Specific outcomes extracted include: study population, location of research, interventions, timeframe and evaluation methods and results. | Global economic evaluations and global meta-analyses where African data is not extractable. |

Pilot review stages were conducted prior to the abstract and full text screening stages. Articles were included if relevant to sub-Saharan Africa AND a stage of pandemic preparedness AND a target CSID AND data systems OR economic analyses (S1 Table). Prior to the abstract screening stage, the complete team of nine reviewers independently screened abstracts from the same 20 articles and discussed discrepancies to improve consistency. Once consensus was reached,

two reviewers independently screened the remaining abstracts for relevance. Articles were excluded if the abstract did not meet the inclusion criteria. Prior to the full-text screening stage, two reviewers screened three sets of ten full-text publications to refine the screening questions, ensuring they were precise and better aligned with the content found across the literature. Once consensus was reached, nine reviewers independently screened the full texts against the full text screening criteria (S2 Fig). Any article deemed relevant at the full text screening stage was screened in duplicate by another reviewer to reach consensus.

## Data charting process

The data charting process was conducted using a data extraction form in Covidence designed by one reviewer and reviewed *a priori* by two reviewers with expertise in the relevant areas. The form was tested on 15 pilot studies by one reviewer, with adjustments made based on the screening results, and further refined before use.

## Data items

The data extracted included article meta-data and thematic content. Article meta-data included year of publication, author institutions, country of data source, source of funding, ethical clearance, and type of publication. Topic data were extracted and thematically categorised based on stage of pandemic preparedness and response framework (S1 Fig), relevance to target CSIDs (Table 1), and type of data system or economic study. A final mapping of challenges and constraints was conducted by assessing topics with the least volume or least relevant literature, and by conducting a narrative synthesis identifying repeated themes describing evidence gaps arising in the final study set.

## Synthesis of results

The PRISMA flow diagram was created using the PRISMA2020 template [27]. We used bibliometric analysis, topic mapping, and narrative summaries to synthesise evidence.

**Geographical distribution of studies.** A bibliometric analysis was conducted by assessing the geographical distribution of the studies across the WHO AFRO region. Temporal trends of authorship were also assessed based on institutional affiliations and publication year, with these results provided in the supplementary information (S3 Fig).

**Climate-sensitive infectious diseases.** The frequency of each CSID was assessed using topic mapping, categorising the proportion of data system or economic studies relevant to each CSID. Studies were also mapped to each stage of pandemic preparedness, with these results provided in the supplementary information (S4 Fig).

**Narrative synthesis.** A narrative synthesis was conducted by categorising data system and economic studies into sub-categories, summarising identified evidence gaps noted by the study authors, and assessing the gender sensitivity of each study. Data system studies were categorised into two sub-categories: 'Design', which included studies that describe the use of multidisciplinary data systems in pandemic preparedness; and 'Usage', which included analyses that use data from multidisciplinary data systems to conduct studies relevant to pandemic preparedness. The categorisation by this method was chosen to account for the large heterogeneity in the data system studies. Economic studies were categorised into three sub-categories based on study type: Economic evaluations, which include analyses that compare costs of interventions with health outcomes, inclusive of CEAs, CUAs, CBAs, whether or not conducted as part of an HTA; costing studies, which include analyses that consider costs without consideration of health outcomes; and priority setting studies, which are more conceptual or commentary based studies that consider the benefit of economic studies for pandemic preparedness.

A gender lens was applied to studies, by assessing the extent that gender was considered across the studies against definitions provided by the WHO intersectionality gender toolkit for research on infectious diseases of poverty [10]: Gender-blind research ignores gender norms, roles, and relations; Gender-sensitive research considers inequality

generated by unequal gender norms, roles, and relations but takes no remedial action to address it; Gender-specific research considers inequality generated by unequal norms, roles, and relations and takes remedial action to address it but does not change underlying power relations. Notably, gender was treated as a binary in this review. Non-binary gender identities were not captured due to limitations of the available literature.

All analyses were conducted in Microsoft Excel [28], and figures were produced in R version 4.3.1 via R studio [29]. The data extracted from included studies can be found at the OSF (https://osf.io/fn8rw/?view_only=5961c57df32741b884fc2181a0da4ceb).

### Protocol amendments

We applied two protocol amendments to our review. First, we refined the fourth abstract screening question relating to data systems by applying more stringent criteria to the data systems theme, as outlined in Table 1. Specifically, we wanted to capture studies that used data from established data systems, that described the creation of new data systems, or that outlined the interoperable use of multi-sectoral data systems. This amendment was required to streamline the screening process conducive to our rapid review timeframe. Second, we did not conduct a pilot of data extraction in duplicate, where two reviewers would independently extract data from the same paper to test the clarity and reproducibility of the form. Instead, the data extraction form was designed by one reviewer and refined by two reviewers with extensive experience in either economics or data systems. The form was then applied to ten studies by an independent reviewer and adjusted based on the results before being finalised for use across the final studies.

## Results

### Selection of sources of evidence

The screening process overview is presented in a PRISMA flow chart (Fig 1). A search across six databases resulted in the identification of 14,252 studies and an additional 112 studies were retrieved from previous reviews conducted by the research team. After removal of 7,059 duplicate studies, 7,305 studies were screened. During screening, 2,255 and 4,774 studies were removed at the title and abstract screening stages respectively. A total of 276 full texts were screened, culminating in 68 studies that were relevant for data extraction. Of the 68 studies, 50 and 18 were categorised as data system and economic studies, respectively.

### Geographical distribution of studies

Data system studies showed broad coverage across the WHO AFRO region, whereas economic studies were much more concentrated in certain countries (Fig 2). Of the data system studies (n = 50), all countries in the WHO AFRO region were represented in at least one study, largely due to three regional-based studies that included all countries [30–32]. Nigeria was the most represented country in data system studies, appearing in 36% (18/50) of studies, followed by Ghana and Kenya, which were each represented in 32% (16/50) of studies. In contrast, of the economic studies (n = 18), nearly half of the WHO AFRO countries were not represented in any studies. South Africa was the most frequently represented country in economic studies, appearing in 28% (5/18) of studies, followed by Ghana and Sierra Leone, each represented in 22% (4/18) of studies. The frequency of publications increased between 2010 and 2023 (S4 Fig), with authorship trends indicating a growing involvement of researchers based in African institutions in CSID research. However, mixed institutional research teams remain dominated by institutions based outside the continent up to 2023.

### Climate-sensitive infectious diseases

From 2010 to 2023, COVID-19, Ebola, Influenza, and Rift Valley fever were the most studied target CSIDs (Fig 3a), with topic mapping revealing broad coverage across all CSIDs in data system studies but much more limited coverage

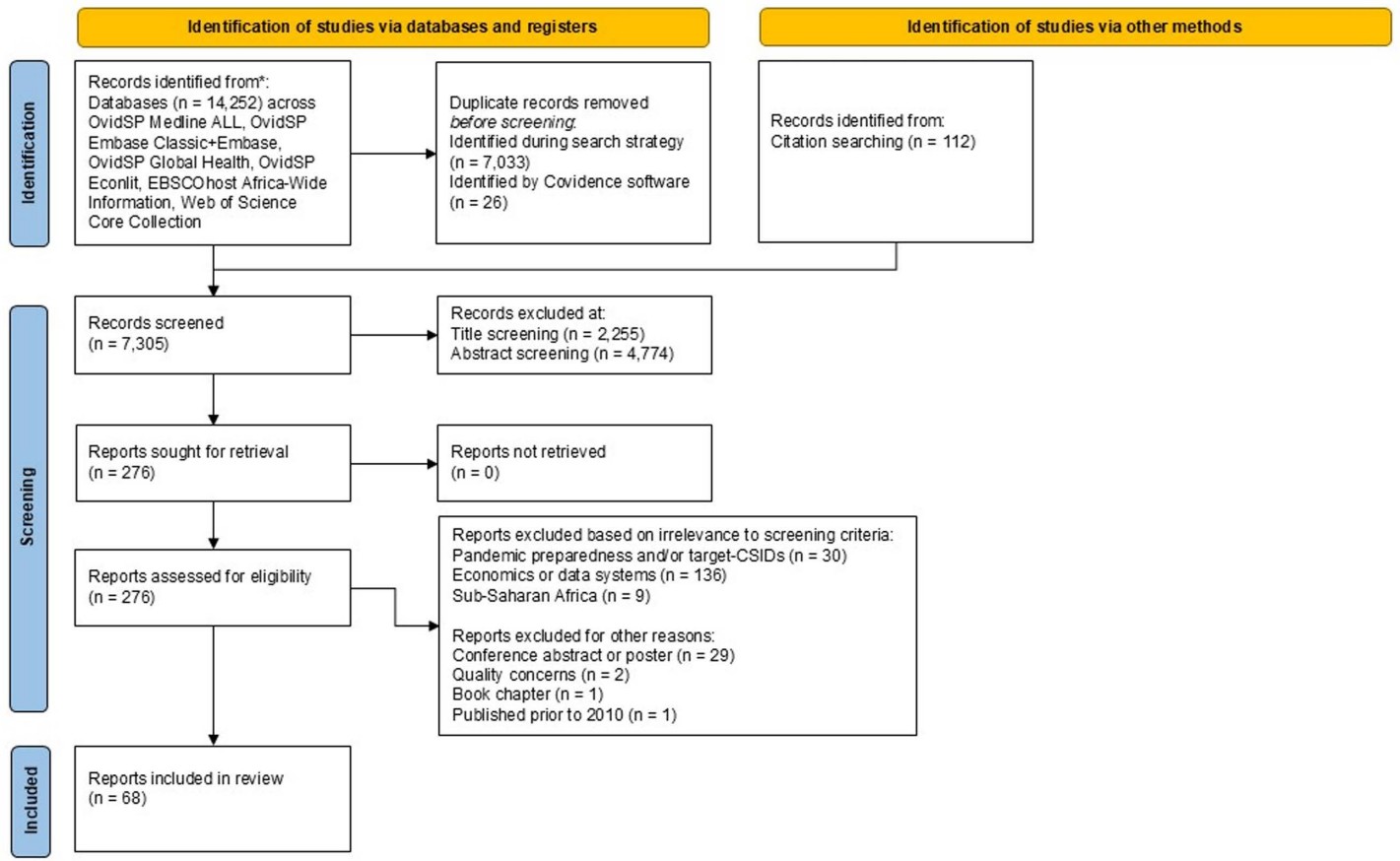

**Fig 1. PRISMA-ScR flow diagram of study selection.**

in economic studies (Fig 3b, 3c). Each CSID was represented at least once in data system studies (Fig3b), though five dominated the literature: Rift Valley fever (22 studies), Ebola virus disease (14 studies), Influenza (12 studies), COVID-19 (12 studies), and Lassa fever (8 studies). In contrast, economic studies were restricted to four CSIDs (Fig 3c), with the majority focusing on COVID-19 (14 studies) and Ebola (6 studies). Influenza and Rift Valley fever were each addressed in a single study, which included Ebola in the same publication [33].

### Narrative synthesis: Data system studies

The following sections provide a narrative summary mapped to the type of study identified through our screening criteria and the evidence gaps identified by the study authors.

The 50 data system studies were categorised into 'Design' studies (n = 19) or 'Usage' studies (n = 33). Design studies described the operational design of data systems for pandemic preparedness. Usage studies were multidisciplinary analyses that integrated data from systems spanning two or more domains across animal, human, or environmental sectors. Three studies were categorised as both Design and Usage studies [34–36].

**Design data system studies.** Of the 19 studies that described data systems in pandemic preparedness (i.e., Design studies), 10 were original research [34,36–44], five were literature reviews [45–49], two were outbreak response case studies [50,51], one was a commentary [52], and one was a letter to the editor [53].

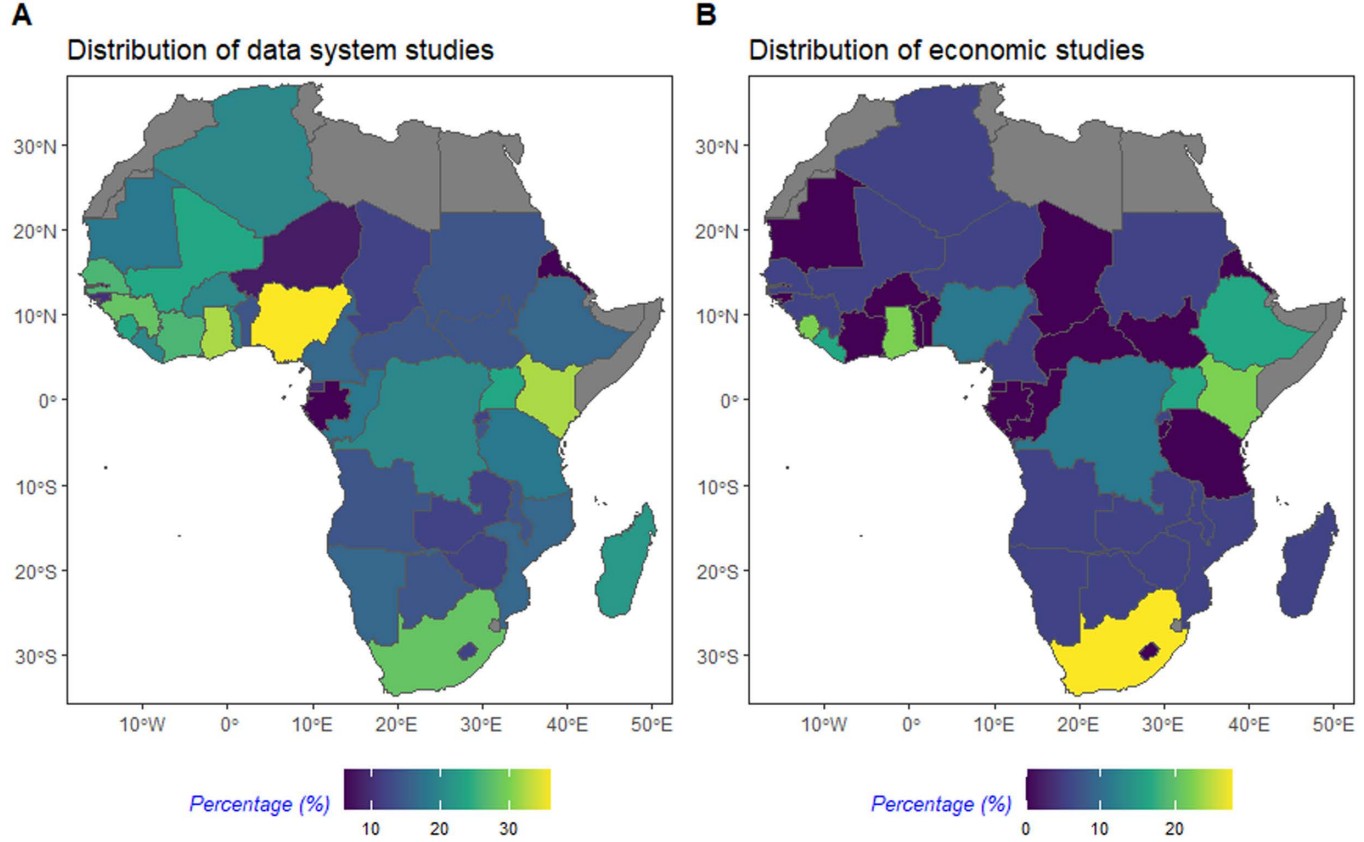

**Fig 2. Distribution of (A) data system studies and (B) economic studies.** The distribution of countries was plotted as percentage of frequency of country included across data system or economic studies by total number of data system or economic studies. Countries coloured in grey correspond to countries outside of the WHO AFRO region. The base layer of the map, showing country borders, was sourced from Natural Earth and accessed using the rnaturalearth package in R Studio. Natural Earth data is in the public domain (https://www.naturalearthdata.com/downloads/).

Across the 19 Design data system studies, 28 data systems were described with international, regional, national, or local scales (Fig 4, S2 Table). Seven of the data systems were surveillance systems integrating human, animal, or environmental domains [37,45–48,51]; six were early warning systems [36,39,42,44,49]; four described community-based surveillance systems [37,40,48,53]; two were mobile surveillance tools used in participatory disease surveillance methods [37,38]; two were communication tools designed to share risk maps [34,49]; two were diagnostic laboratory systems [46,50]; two were online genomic sequencing systems [35,52]; two were systems to assess One Health capacity of countries [43] or coordinate a pandemic response [50]; and one study focused on off-line software designed to predict the causative pathogen of an outbreak [41]. The varying scopes of these data systems highlight the range of data sharing for CSID outbreak detection and response, spanning local [40,50], national [36,42,44,47,50,51,53], regional [37,38,43,45,48], and international levels [34,35,37,39,41,46,49,52].

Evidence gaps described by Design data system study authors included limited data availability or sharing across sectors, and surveillance capacity. These limitations encompassed factors such as funding [38,50] and data availability or reporting issues within the animal sector [42,47]. In the human sector, delays in reporting were often attributed to paper-based data transmission and inadequate infrastructure [37,42,44,49]. In the environmental sector, an absence of local stakeholder involvement was reported, which resulted in environmental monitoring being an isolated academic exercise

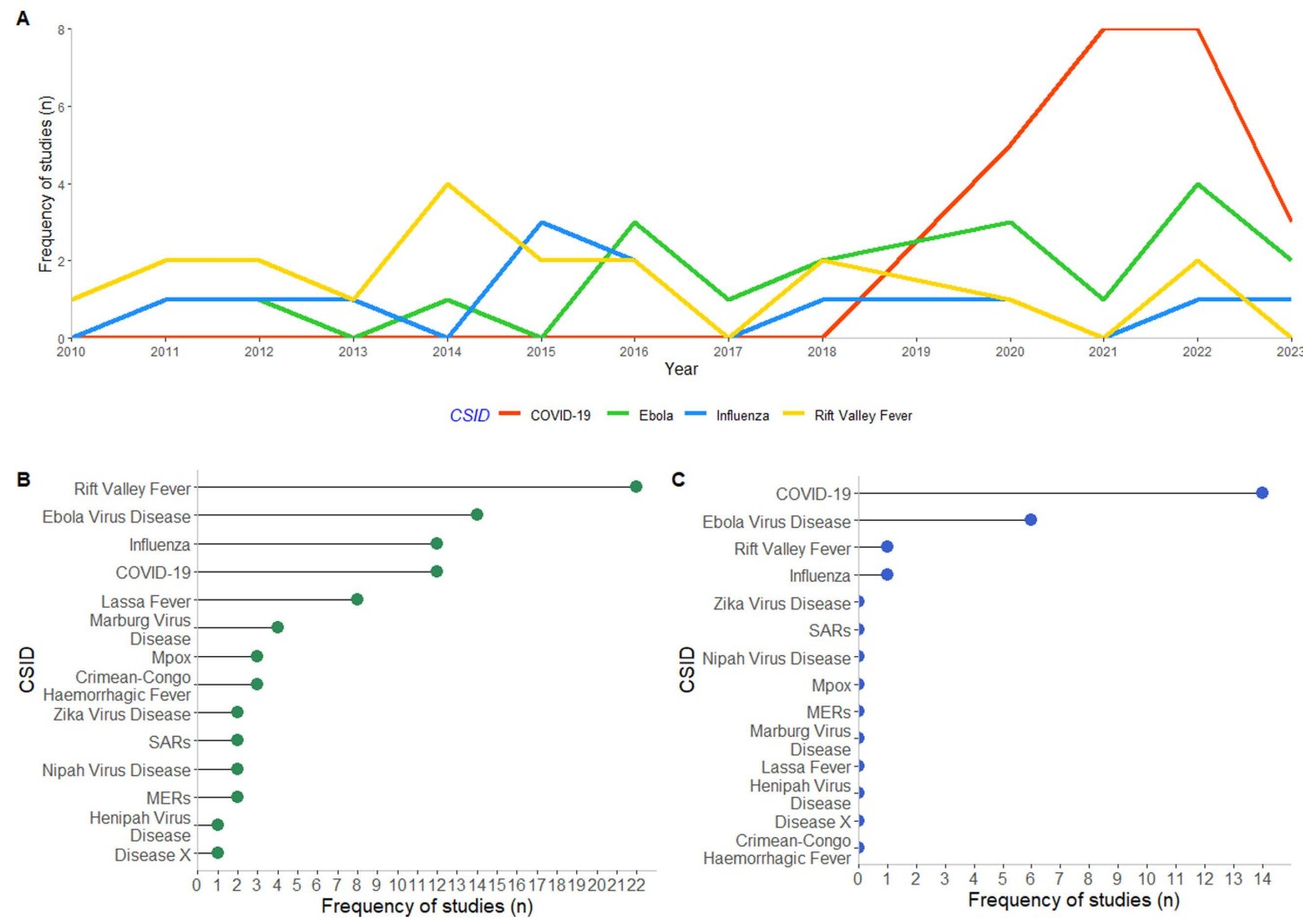

**Fig 3. Frequency of most frequently cited CSID over publication period (A), and frequency of all CSIDs cited across (B) data system and (C) economic studies.** Studies could be mapped to more than one CSID. Abbreviations: COVID-19: Coronavirus Disease 2019; SARS: Severe Acute Respiratory Syndrome; MERs: Middle East Respiratory Syndrome.

driven by national and international stakeholders [38]. Collaborative efforts across the human, animal, and environmental sectors were hindered by the absence of standardised guidelines, policies, or frameworks [43,48].

Evidence needs described by Design data system study authors encompassed inclusion of vulnerable groups through digitally enabled participatory methods and increased cross-sectoral data sharing. Specifically, these needs encompass the integration of vulnerable groups into disease surveillance [37,47,52,53] or control efforts [38] through adoption of participatory methods harnessing mobile phone technologies [42,47]. Additionally, studies advocated for incorporation of climate data into analyses to enhance understanding of CSID transmission and development of early warning systems [39], or long-term planning of response, control, and mitigation strategies [2]. Furthermore, studies described a need for greater political will and capacity to facilitate data-sharing across human and animal sectors [35,37,45,47,52,53] underpinned by established data standards, protocols, or centralised platforms [37,42–45,48,49,53], the pooling of multisectoral funds to support One Health policies [45], and increased private sector [50] or governmental [38] financial support.

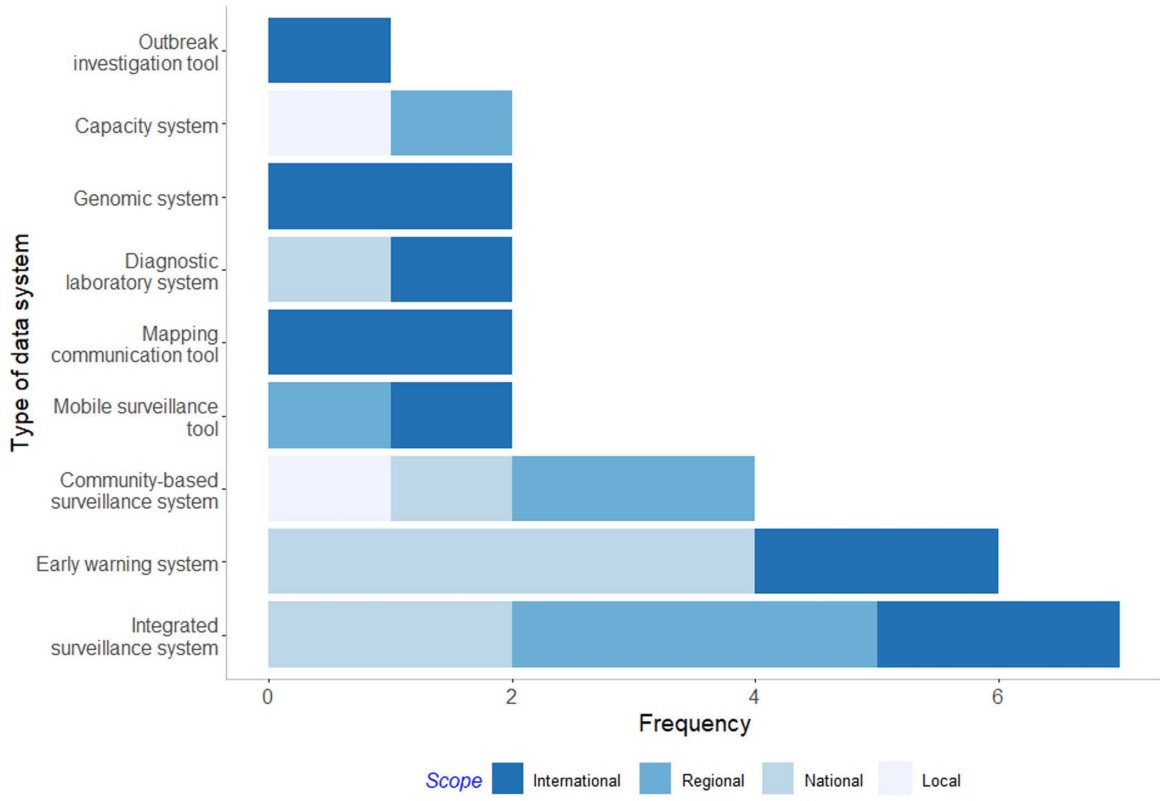

**Fig 4. Frequency of Design data systems by type and scope.**

**Usage data system studies.** Of the 33 multidisciplinary studies that integrated data from data systems (i.e., Usage studies), 12 were risk mapping studies [32,34,36,54–62], ten were association studies [63–72], seven were both risk mapping and association studies [2,31,73–77], and four were phylogenetic analyses [30,35,78,79] (Fig 5, S3 Table). We did not find any attribution studies. The different study types included varying domains across human, animal, and environmental data. For example, most association studies (n = 7) investigated the association of environmental variables on human response variables, such as that of ambient temperature exposure on COVID-19 transmission, whereas all phylogenetic analyses (n = 4) investigated spill-over between humans and animals.

Evidence gaps described by Usage data system study authors surrounded data availability and cross-sector data sharing or coordination. Specifically, gaps revolved around inadequate availability of animal health data, often stemming from weak surveillance across the animal sector [54,55,66,74,75] or spatial biases in sampling and surveillance data [2,30,59,78,79]. Additionally, deficiencies were noted in coordination between human, animal, and environmental surveillance systems [58] compounded by limitations in reporting capacity [35,72] and limited temporal and spatial resolution of selected environmental data [31].

Evidence needs described by Usage data system study authors included increased empirical data collection during outbreaks [54,55] or routine surveillance [66] supported by participatory approaches to ensure data collection and inclusion of vulnerable groups [58]. Furthermore, there was a call for the adoption of an integrated, One Health approach to surveillance, emphasising cross-sectoral and transboundary collaboration [35] alongside improved data sharing across animal and human sectors [30,62,78]. Such initiatives should be supported by established data standards [63]. Additionally, studies recognised the need for human and animal data linkage on a fine spatial scale [54,55,70,79], coupled with

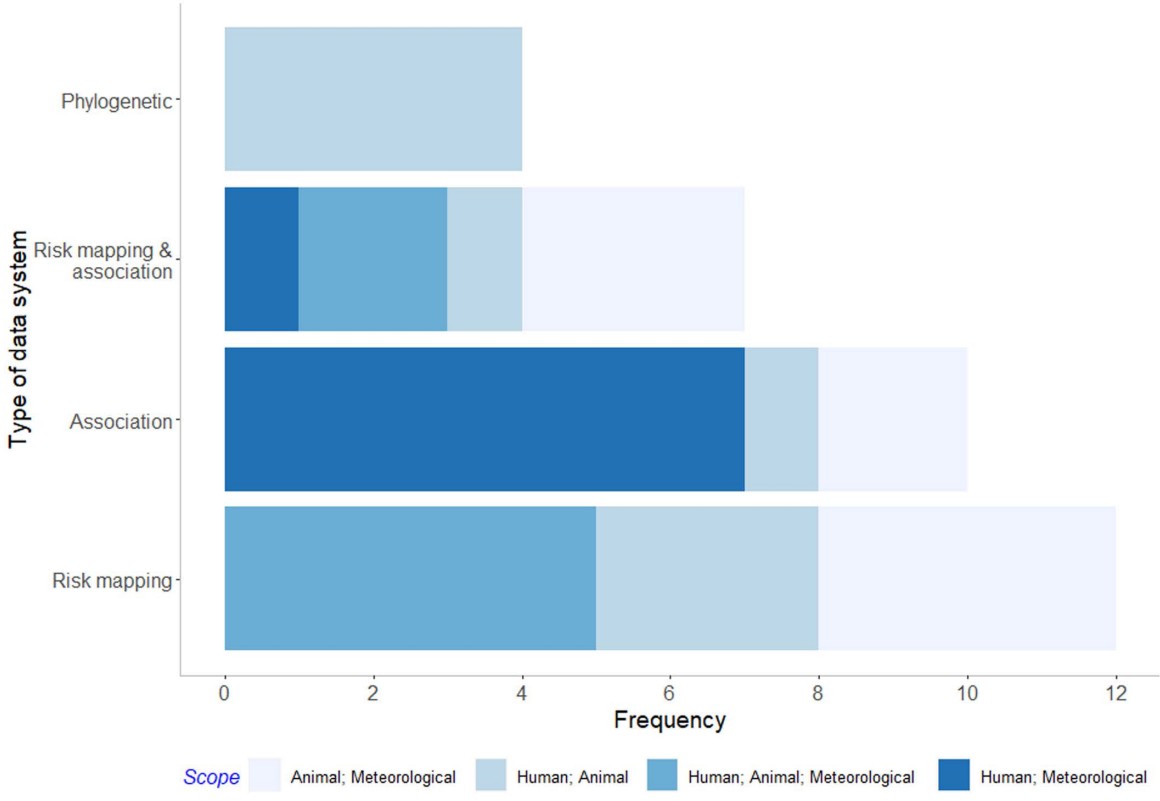

**Fig 5. Frequency of Usage data systems study by study type and integration of data domain.**

the integration of environmental data to deepen our understanding of the impact of climate change on CSID transmission [2,59,60,65,68,76]. These insights would inform the allocation of surveillance or control strategies [31,60,64,77].

**Gender lens of data system studies.** Of the data system studies that included human data (n = 41), 38 studies were considered gender-blind, two were considered gender-sensitive [41,44], and one gender-specific [40]. Of the gender-sensitive studies, one described a system for predicting the causative pathogen of an outbreak which was programmed to disaggregate by gender and age [41], and one study reported Rift Valley fever cases by gender [44]. The authors of the gender-specific study included women as key stakeholders in a surveillance data system, acknowledging their important role in disease detection despite their frequent under-representation [40]. This study addresses the evidence need for participatory approaches to ensure the inclusion of vulnerable groups in data collection (identified in reference 58).

### Narrative synthesis: Economic studies

The 18 economic studies were comprised of economic evaluations (n = 10), which included CEAs and CUAs; costing studies (n = 3); and priority setting papers (n = 5).

**Economic evaluation studies.** Economic evaluation studies (n = 10) investigated the cost effectiveness of vaccination strategies [80–84], non-pharmaceutical interventions (NPIs) [85–87], or clinical critical care treatments [88,89], and were limited to COVID-19 (n = 8) or Ebola (n = 2) (Table 2).

Evidence gaps described by economic evaluation study authors highlighted significant deficiencies in data availability in relation to intervention efficacy and resource use/cost information, and limited regional capacity for the conduct of economic evaluations. Specifically, studies reported a lack of data availability surrounding COVID-19 vaccination

**Table 2. Summary of economic evaluation studies.**

| First Author | Year of publication | CSID(s) | Population considered | Intervention/ comparators | Uncertainty analysis | Gender-lens |
|---|---|---|---|---|---|---|
| Orangi [84] | 2022 | COVID-19 | Population of Kenya | Vaccination roll-out strategies | Univariate and probabilistic sensitivity analysis | Gender-blind research |
| Reddy [85] | 2021 | COVID-19 | Population of KwaZulu-Natal | Combinations of five NPIs across R = 1.5 and 1.2 epidemiological scenarios | One-way and multiway sensitivity analysis | Gender-blind research |
| Reddy [80] | 2021 | COVID-19 | Population of South Africa | Vax supply and vaccination pace strategies among Re = 1.4 and 2-wave scenarios | One-way and multi-way sensitivity analysis | Gender-blind research |
| Asamoah [86] | 2020 | COVID-19 | COVID-19 cases in Ghana | Combinations of five NPIs | Varied estimated parameter values to formulate parameters that would generate 'optimal control model' | Gender-blind research |
| Ruiz [81] | 2023 | COVID-19 | Population of Nigeria | Vax types, viral vectors, delivery method, prioritisation groups | Analysis conducted over a range of scenarios and presented as main results | Gender-blind research |
| Liu [82] | 2022 | COVID-19 | Population across 27 African countries | Vaccination (viral vector vaxs and mRNA vaxs), and no vaccination | Sensitivity analysis | Gender-blind research |
| Obeng-Kusi [83] | 2022 | Ebola | 2014-2016 EVD cases in DRC, Liberia, Sierra Leone, Uganda | Vax package (vax, storage, maintenance, and administration) | Probabilistic sensitivity analyses | Gender-blind research |
| Kairu [88] | 2021 | COVID-19 | Hospitalised COVID-19 patients admitted between March 2020 and January 2021 | Essential Care, Advanced Critical Care, or maintaining status quo | One-way sensitivity analysis and a probabilistic sensitivity analysis | Gender-blind research |
| Beshah [89] | 2023 | COVID-19 | COVID-19 patients (age 18+) enrolled at treatment centre and home-based isolation care between Jan 1 - May 31, 2021 | Non-invasive and invasive critical case management | Probabilistic and one-way sensitivity analysis | Gender-blind research |
| Kellerborg [87] | 2020 | Ebola | EVD patients from 2014-2016 Sierra Leone outbreak | Time period of interventions taking place (original time or 4 weeks earlier) | Univariate sensitivity analysis | Gender-blind research |

Abbreviations: CEA: Cost-effectiveness analysis; Vax: Vaccine.

effectiveness [84], COVID-19 immunity data [80–82,84], and costing data [84,85]. Some studies noted a lack of local data regarding COVID-19, necessitating reliance on UK estimates [84] or rendering the models unparameterisable to the local setting [88]. Others discussed limited regional capacity for economic evaluations, specifically highlighting such aspects as an absence of established willingness to pay thresholds in South Africa [80] or more generally across low and middle-income countries (LMICs) [83]; a limited ability to compare cost-effectiveness estimates due to a lack of economic evaluations [89]; or a failure to consider health system restraints [81]. Additionally, one study addressed the challenge of incorporating human productivity loss into economic analyses of Ebola [87].

Evidence needs described by economic evaluation study authors included broadening cost-effectiveness analyses of vaccines to incorporate the far-reaching impacts of COVID-19 [84] and accounting for wealth inequity through subpopulation analyses [83]. Additionally, there was a need for extrapolation of epidemiological and economic models to other

settings for informing vaccination purchasing [81], and incorporation of timing into evaluations for more realistic cost-effectiveness assessments [82].

**Costing studies.** The three costing studies analysed the costs of various initiatives across vaccination programs, a surveillance system, and outbreak response (Table 3 ). One study presented cost estimates of a COVID-19 vaccination program in Ghana utilising the COVID-19 Vaccination Introduction and Deployment Costing Tool under the leadership of the Ghana HTA program [90]; another was a micro-costing study of a community-based surveillance (CBS) system for detecting Ebola and COVID-19 (among non-target CSIDs) in Sierra Leone [91]; and the final study costed the US CDC's response to Ebola outbreaks across Sierra Leone, Guinea, and Liberia [92].

Evidence gaps described by costing study authors included delays in reporting by community health workers within the CBS and inconsistency in performance incentives being distributed [91], and lack of cost records attained within Ebola outbreaks resulting in the potential underestimate of costs [92].

Evidence needs described by costing study authors included using the established COVID-19 Vaccine Introduction and Deployment Costing (CVIC) tool for costing the introduction of vaccinations across LMICs [90] and incorporating reporting of symptoms of COVID-19 and other future emerging infections into existing CBS for early-warning systems [91].

**Priority setting studies.** Priority setting papers described the role of economic evaluations and associated priority setting tools or guidelines to support decision-making (Table 4). Among the five studies, one described the application of HTAs for equitable decision-making in COVID-19 in LMICs [93], while another described the Evidence-to-Decision Framework utilised by the South African Grading of Recommendations Assessment, Development and Evaluation (GRADE) network to generate 42 rapid reviews for timely COVID-19 response [94]. Another study considered economic evaluation, community participation, and ethical considerations in priority setting for COVID-19 response [95], and one discussed economic evaluation and evidence synthesis processes for priority setting across the four stages of pandemic preparedness [96]. Lastly, a study discussed potentially cost-effective control strategies for animal and zoonotic diseases, including Rift Valley fever, Influenza and Ebola, in pastoralist populations [33], marking the sole economic study to address CSIDs beyond Ebola or COVID-19.

Evidence gaps described by priority setting study authors included potential barriers to the application of HTAs and challenges stemming from insufficient collaboration or capacity within the animal sector. Specifically, these gaps encompassed the complexity of HTAs as a barrier to decision-making [93,96]; a lack of collaboration resulting in duplicated efforts and resource wastage during the COVID-19 pandemic [94]; limitations in animal disease surveillance and data maintenance impeding timely outbreak detection; and the absence of a stringent framework for monitoring the success of animal interventions leading to hesitancy in investment by institutional donors [33].

Evidence needs described by priority setting study authors included increasing regional capacity and collaboration while utilising participatory methods to achieve equitable resource allocation. This would include leveraging strong leadership with

**Table 3. Summary of costing studies.**

| First Author (Reference) | Year of publication | CSID(s) | Population considered | Intervention/comparators | Uncertainty analysis | Gender-lens |
|---|---|---|---|---|---|---|
| Nonvignon [90] | 2022 | COVID-19 | All Ghanaians aged 16 years and above and are not pregnant | Vaccination program | Multi-way sensitivity analysis | Gender-blind research |
| Mergenthaler [91] | 2023 | Ebola; COVID-19 | Sierra Leone health system (Micro-costing study from a health system perspective) | CBS as part of the electronic IDSR system | NR | Gender-blind research |
| Carias [92] | 2018 | Ebola | Three Ebola outbreak clusters from 2016 in Sierra Leone, Guinea, and Liberia/Somalia | United States CDC's response to outbreaks. | NR | Gender-blind research |

Abbreviations: CBS: Community-based surveillance; IDSR: Integrated Disease Surveillance and Response.

**Table 4. Summary of priority setting studies.**

| First Author | Year of publication | Aims | CSID(s) | Relevance to priority setting | Gender-lens |
|---|---|---|---|---|---|
| Anantha-krishnan [93] | 2022 | Discuss examples of LMICs using HTAs during COVID-19 response and discuss broader HTA | COVID-19 | Advocates for HTA use in pandemic response through case studies across LMICs | Gender-specific research |
| Mosam [95] | 2020 | Discuss economic evaluation, community participation, and ethical considerations in priority setting of COVID-19 response in South Africa | COVID-19 | Advocates for development of reliable but rapid processes for resource allocation during pandemic preparedness, highlighting importance of economic evaluation | Gender-blind research |
| Kapiriri [96] | 2022 | Discuss how priority setting and resource allocation could be integrated into WHO pandemic preparedness framework to inform COVID-19 pandemic response, with Ebola examples | Ebola; COVID-19 | Provides a framework for incorporating economics across the WHO pandemic preparedness stages (pre-epidemic preparedness, alert phase, control phase, evaluation phase) | Gender-specific research |
| McCaul [94] | 2022 | Discuss COVID-END body utilising South Africa case study | COVID-19 | Describes COVID-END, a global initiative of 50 evidence synthesis or support groups for generation and communication of trustworthy, rapid, and equitable evidence syntheses to inform clinical and public health decisions and vaccination rollouts | Gender-blind research |
| Zinsstag [33] | 2016 | Address options for cost-effective control of animal disease and zoonoses in pastoral areas, as well as for disease surveillance and the financing of animal health services, utilising case studies in Ethiopia and Eastern/Western Africa | Influenza; Ebola; Rift Valley fever | Discusses frameworks for evaluating economic efficiency of animal disease control across stages of pandemic preparedness, with a particular focus on surveillance | Gender-blind research |

Abbreviations: LMIC: Low and Middle-Income Country; HTA: Health Technology Assessment; COVID-END: COVID-19 Evidence Network to support Decision-making.

political and institutional support to integrate HTAs into policy-making and foster regional technical expertise [93] and facilitating collaboration between designated working groups to prevent duplication of evidence synthesis efforts (30). Additionally, authors suggested employing mobile phone technologies in near real-time community-based surveillance systems, integrating multi-disease zoonotic CSID control approaches to optimise intervention benefit-cost ratios in pastoral areas [33], and enhancing community participation and expert ethical consideration for equitable resource allocation during pandemics [95,96].

**Gender lens of economic studies.** Of the economic studies, 16 were considered gender-blind according to the WHO intersectionality toolkit [10], and two were gender-specific. Of the gender-specific studies, one study discussed gender and socioeconomically marginalised groups as stakeholders that could benefit from HTAs, noting that power dynamics between groups could limit their inclusion into HTA processes [93]. The other gender-specific study discussed how vulnerability to pandemics could be exacerbated by gender and socioeconomic status, highlighting the importance of their consideration into research priorities and resource allocation within pandemic preparedness [96].

## Critical appraisal within sources of evidence

A critical appraisal of evidence was not conducted.

## Discussion

We conducted a rapid scoping review to identify research needs for improving priority setting for pandemic preparedness in sub-Saharan Africa, focusing on data system and economic studies related to 14 CSIDs with pandemic potential.

Peer-reviewed economic studies on CSIDs were limited with most research focusing on COVID-19 and Ebola and originating from South Africa, highlighting a gap in resource allocation and capacity for other diseases. The high degree of heterogeneity across data system studies, along with poor data sharing across siloed sectors, indicates a need for clear data system definitions and guidelines to develop interoperable and scalable solutions. Additionally, the limited inclusion of gender-sensitive perspectives in both economic and data system studies points to a significant gap that should be addressed to ensure more comprehensive and equitable pandemic preparedness strategies. Taken together, these identified gaps present important opportunities to enhance priority setting, improve resource allocation, and achieve more equitable health outcomes across the region.

## Gaps and opportunities

The limited number of economic studies, along with their concentration on COVID-19 and Ebola and primarily conducted in South Africa, suggests a broader deficiency in capacity for conducting similar evaluations across the rest of the WHO AFRO region and for other CSIDs. This observation is consistent with findings from a previous review that assessed HTA institutionalisation in sub-Saharan Africa [97], which highlighted a lack of technical expertise and tools for context-specific decision-making aligned with locally agreed frameworks for economic evaluation.

To address these deficiencies, developing national HTA frameworks that include the use of context specific economic evaluations could significantly enhance priority setting systems [18,98]. These frameworks should incorporate a diverse range of stakeholder perspectives and extend the usual role of HTAs to address key questions, such as how to fund or create priority setting processes that link climate change and CSID response, and what existing structures support intersectoral actions for pandemic preparedness. Such country-specific HTA frameworks would help assess existing resources and could be applied to a range of CSIDs beyond COVID-19 and Ebola, including emerging threats such as Mpox, which was recently declared a PHEIC by the WHO [99].

We found a large degree of heterogeneity in data system literature, reflecting a lack of clear and universally agreed-upon definitions of data systems related to pandemic preparedness. Establishing these definitions would guide the creation of more effective data systems that provide decision-makers with accurate and timely information. For pandemic preparedness against CSIDs, these definitions should emphasise interoperability and data harmonisation to enable the routine integration of diverse data sources, including human health, animal health, and environmental data.

Such data systems would enhance data sharing across traditionally siloed sectors, addressing a significant gap identified by both data system and economic studies in this review. Sectoral silos and a lack of digital integration platforms are common barriers to data sharing, which hinder evidence-driven decision-making in a One Health context [100,101], despite recommendations from the One Health Joint Plan of Action [101]. The Digital One Health (DOH) framework, based on the FAIR principles (Findability, Accessibility, Interoperability, and Re-use), aims to address these barriers by consolidating data-sharing across five key pillars [102]. These pillars focus on harmonising and automating standardisation according to ethical and legal guidelines, determining which data can be shared, such as pathogen characteristics and patient gender, or withheld, such as patient name or contact. The framework is currently being piloted in antimicrobial resistance surveillance in Uganda [100]. Tailoring such frameworks to the specific challenges faced across decision-making in pandemic preparedness could bridge siloed efforts, enhance coordination, and provide high-quality data for pandemic preparedness decision-making.

Limited gender-sensitivity across both data system and economic studies may result in inadequate consideration of gendered impacts and needs in pandemic preparedness, despite a growing body of evidence that highlights the differential effects of pandemics on various genders [7–10]. Participatory approaches can provide methodologies for gender-transformative change, by identifying vulnerable groups that are context specific and including them into research and subsequent decision-making [10]. Digital participatory methods, such as use of mobile phone applications for reporting animal cases can facilitate near-real-time reporting of suspected outbreaks [103], and also help to incorporate

hard-to-reach populations into decision-making processes. Importantly, digital inequities may hinder the inclusion of marginalised groups, particularly women and girls [9,103], due to limited access to technological education [104], especially in LMICs. Therefore, developing participatory digital tools that are culturally and contextually informed and investing in digital skills for those who need them most, should be an important consideration for pandemic preparedness.

We did not find any data system studies that directly attributed human CSID-related health outcomes to human-caused climate change, despite numerous studies assessing associations between environmental variables and health outcomes. This reflects a broader gap in attribution research; a recent report identified only 13 studies globally since 2013 that rigorously assessed health impacts attributable to human-caused climate change [105], with the only CSID study that attributed human-caused climate change to increased childhood malaria in sub-Saharan Africa [106]. The lack of attribution research on CSIDs is notable, given that the impact of climate change on CSID-related health outcomes is a priority for attribution research [105]. Enhancing data integration across human, animal, and environmental domains through interoperable systems could facilitate assessments of long-term trends and associations, inform counterfactual analyses of climate scenarios, and support economic evaluations of mitigation and adaptation strategies [107]. Such efforts could significantly strengthen the evidence base on the health impacts of climate change on CSIDs.

## Limitations

The rapid scoping review approach, while allowing for a timely synthesis of evidence, introduced several limitations. Due to the expedited timeline, we did not conduct a grey literature search, which may have led to the omission of relevant sources such as governmental policy reports describing data system use and economic evaluations. To mitigate this, we leveraged the expertise of the project steering committee, who identified additional relevant studies based on their experience across complementary reviews. A single unified search strategy was used to identify data system and economic studies, which may have affected the breadth of our scoping search; however, we are confident that the trends identified in the review are consistent with recent literature, as noted in the Discussion section. The rapid nature of the review also precluded in-depth critical appraisal of included studies, meaning that while we provide a broad overview of the literature, future reviews would be improved by incorporating comprehensive appraisals to better evaluate the reliability and validity of the evidence base. Additionally, we limited our search to English-language publications which may have introduced an English-language bias, potentially excluding studies from regions where English is not the primary language and influencing our conclusions.

To manage the number of publications, we only included studies that explicitly mentioned target CSIDs. This may have excluded some publications relevant to pandemic preparedness of CSIDs in less explicit yet important ways, such as focusing on the mode of transmission rather than the pathogen or illness. Broadening the scope to include these studies may provide useful insights into wider data systems or economic evaluations for pandemic preparedness, such as the use of Natural Capital Accounting [108] to assess the broader impacts of disease outbreaks on ecosystem services and public health.

The gender analysis treated gender as a binary (man/woman) due to limitations in the available literature. This may overlook the experiences of individuals who identify outside the binary, and future research should adopt more inclusive definitions of gender to better capture the full spectrum of gender identities.

Finally, neither the lead nor senior author is African (although latter is African resident), and we recognise that interpretations may be shaped by training, institutional affiliations, and lived experiences. To address this, we drew on the expertise of a multidisciplinary, international research and authorship team predominantly composed of African researchers. All authors collaboratively contributed to the review design, analysis, and interpretations, including on gender equity components.

## Conclusions

Our rapid scoping review revealed significant evidence gaps in data systems and economics for priority setting of pandemic preparedness in sub-Saharan Africa. Key issues include limited African-led research, a scarcity of peer-reviewed

economic studies on most CSIDs, inadequate data sharing across human, animal, and environmental domains, and insufficient application of gender-sensitive perspectives. Addressing these gaps presents important opportunities to enhance decision-making in pandemic preparedness and ensure more effective and inclusive responses to emerging infectious disease threats. This need is especially urgent given the changing climate, which increases the risk of CSID pandemics, as demonstrated by the recent Mpox outbreaks.

## Supporting information

**S1 Fig. Pandemic preparedness framework for climate sensitive infectious disease in Africa.** Adapted from: World Health Organization. (2014). Ebola and Marburg virus disease epidemics: Preparedness, alert, control, and evaluation. https://www.who.int/publications-detail-redirect/WHO-HSE-PED-CED-2014.05
(DOCX)

**S2 Fig. Full text screening criteria.**
(DOCX)

**S3 Fig. Percentage of data system or economic studies mapped to stages of pandemic preparedness.** Studies could be relevant to all four stages; percentage was calculated as tagged stage by total of tagged stages in data system studies (n = 50) or economic studies (n = 18).
(DOCX)

**S4 Fig. Number of studies published over time, by geographical distribution of author affiliations.** The institutional affiliations of authors were categorised as follows: African - all author affiliations located within Africa; African & International - author affiliations located both within Africa and internationally; and International - all author affiliations located outside of Africa.
(DOCX)

**S1 Table. Screening guidance.**
(DOCX)

**S2 Table. Summary of Design Data system studies.**
(DOCX)

**S3 Table. Summary of Usage Data System studies.**
(DOCX)

## Author contributions

**Conceptualization:** Ellie A. Delight, Ariel A. Brunn, Francis Ruiz, Yang Liu, Bubacarr Bah, Bernard Bett, Benjamin Uzochukwu, Oladeji K. Oloko, Esther Njuguna, Kris A. Murray.

**Data curation:** Ellie A. Delight, Jessica Gerard, Jane Falconer.

**Formal analysis:** Ellie A. Delight, Jessica Gerard.

**Funding acquisition:** Ariel A. Brunn, Kris A. Murray.

**Investigation:** Ellie A. Delight, Ariel A. Brunn, Francis Ruiz, Jessica Gerard, Jane Falconer, Yang Liu, Bubacarr Bah, Bernard Bett, Benjamin Uzochukwu, Kris A. Murray.

**Methodology:** Ellie A. Delight, Ariel A. Brunn, Francis Ruiz, Jane Falconer, Kris A. Murray.

**Project administration:** Ariel A. Brunn, Kris A. Murray.

**Resources:** Ariel A. Brunn, Francis Ruiz.

**Software:** Jane Falconer.

**Supervision:** Ariel A. Brunn, Kris A. Murray.

**Validation:** Ariel A. Brunn, Jane Falconer, Kris A. Murray.

**Visualization:** Ellie A. Delight, Jessica Gerard.

**Writing – original draft:** Ellie A. Delight, Ariel A. Brunn, Francis Ruiz, Jessica Gerard.

**Writing – review & editing:** Ariel A. Brunn, Francis Ruiz, Jessica Gerard, Jane Falconer, Yang Liu, Bubacarr Bah, Bernard Bett, Benjamin Uzochukwu, Oladeji K. Oloko, Esther Njuguna, Kris A. Murray.

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
