## [Decision Letter · Decision Letter 0]

13 Nov 2024

PGPH-D-24-02054

Gaps and Opportunities for Data Systems and Economics to Support Priority Setting for Climate-Sensitive Infectious Diseases in Sub-Saharan Africa: A Rapid Scoping Review

Dear Dr. Delight,

Thank you for submitting your manuscript to PLOS Global Public Health. After careful consideration, we feel that it has merit but does not fully meet PLOS Global Public Health’s publication criteria as it currently stands. Therefore, we invite you to submit a revised version of the manuscript that addresses the points raised during the review process.

We look forward to receiving your revised manuscript.

Kind regards,

Charin Modchang, Ph.D.

Academic Editor

Journal Requirements:

1. Figure 3: please (a) provide a direct link to the base layer of the map (i.e., the country or region border shape) and ensure this is also included in the figure legend; and (b) provide a link to the terms of use / license information for the base layer image or shapefile. We cannot publish proprietary or copyrighted maps (e.g. Google Maps, Mapquest) and the terms of use for your map base layer must be compatible with our CC-BY 4.0 license. 

Additional Editor Comments (if provided):

We have received comments from the reviewers. Please kindly provide a detailed response to their comments.

Reviewers' comments:

Reviewer's Responses to Questions

**Comments to the Author**

1. Does this manuscript meet PLOS Global Public Health’s publication criteria ? Is the manuscript technically sound, and do the data support the conclusions? The manuscript must describe methodologically and ethically rigorous research with conclusions that are appropriately drawn based on the data presented.

Reviewer #1: Yes

Reviewer #2: Yes

2. Has the statistical analysis been performed appropriately and rigorously?

Reviewer #1: Yes

Reviewer #2: N/A

3. Have the authors made all data underlying the findings in their manuscript fully available (please refer to the Data Availability Statement at the start of the manuscript PDF file)?

Reviewer #1: Yes

Reviewer #2: Yes

4. Is the manuscript presented in an intelligible fashion and written in standard English?

Reviewer #1: Yes

Reviewer #2: Yes

5. Review Comments to the Author

Reviewer #1: Major comments:

Given that the first objective is to “map key themes and entry points for future African-led research,” it would be helpful to have a positionality statement (or equivalent). If the lead author(s) are not African, some reflection on this is needed.

This is an extremely ambitious project that deals with a timely and high priority issue. However, I’m left feeling like it’s trying to do “too much.” The narrative review would be more than sufficient as a standalone paper, so adding in the bibliometric analysis and the thematic mapping (and even possibly the gender analysis, which didn’t seem truly intersectional to me) (a) makes it a bit difficult to grasp the study’s key objectives, and (b) gives insufficient attention to these important elements. I would suggest significantly narrowing the scope of this review, or at minimum move some content to appendices.

Some more justification on why COVID-19 was included as a CSID is needed. I understand initial spillover was (almost certainly) zoonotic, but I’m assuming all of the literature you reviewed dealt with subsequent person-to-person transmission, which is not really climate-sensitive.

Minor comments:

Abstract

A summary of your exclusion criteria would be helpful. As it currently stands, I can’t quite understand the scope of your review.

Introduction

Line 148: “we extended a conceptual framework.” Is this a conceptual framework that already exists and you extended it? Or something new you developed?

Line 161: I don’t think intersectional approaches to analyses necessarily imply representation, or vice versa. Suggest clarifying language here

Line 167-179: This reads like you’re trying to wrap quite a lot into a single project, and it makes it quite difficult for the reader to parse. I would harmonize these into a single list, and add clarity however you can (e.g., shorter sentences, a conceptual diagram, etc.).

Methods

Table 1: with AI tools (e.g., Google Translate and DeepL), it’s a bit harder to justify exclusion of articles based on language. Some text to justify this (or discuss it as a limitation) is needed. Also, the climate change inclusion criteria cell is really confusing as written—for instance you say articles that referred to infectious diseases impacted by climate change were included, but then you list variables defining climate change, not infectious diseases. Did you include all CSIDs with zoonotic potential, or just those with pandemic potential? Ideally this would be in list form (true for the whole table).

Line 231: Independently screened the remaining abstracts?

Line 273: “Use data from data systems.” Was this intended to say “from multidisciplinary data systems”?

Lines 284-291: Is intersectionality relevant to all of the study types listed in the previous paragraphs? I guess I’m not sure how this makes sense for studies that fit the data systems: design category since those aren’t research studies.

Line 302: “the large volume of literature utilizing data from data systems.” I’ll say! Perhaps you want to be more specific here?

Throughout: I notice you use the term “iteratively refine” a few times. It’d be helpful to have just a few extra words in these locations detailing how this was done, or at minimum what was the endpoint.

Results:

I’d suggest breaking down the results for Design data systems into subsections with subheaders, to make it easier to follow

Lines 498-513: this seems to be more of a gender analysis than analysis of intersectionality. What about other marginalized identities, besides gender? Also, I assume all of these studies treated gender as a binary, but it’d be worthwhile to reflect on that.

Line 522: I am surprised that you only found 8 studies investigating the cost effectiveness of COVID-19 measures in sub-Saharan Africa, and only 2 for Ebola also seems really small. Can you elaborate on this—where (in terms of your exclusion criteria) did most of these get weeded out?

Lines 573-586: I’m surprised it was pretty much all Ebola and COVID-19, especially as COVID-19 isn’t really a CSID (on this note, I question the decision to include it since it dominates so much of your review). It’d be helpful to see a list of the diseases/pathogens you included—I assume vectorborne diseases were excluded?

Reviewer #2: The paper provides an overview of research and points towards research gaps for a pertinent issue: the role of data systems and economics in decision-making for climate-sensitive diseases. The paper is generally well-written and presented. Some comments to improve the manuscript further:

1. line 71, p.3: Temperature and rainfall are not climate hazards. You can rephrase this to "...the frequency and intensity of climate hazards (e.g., droughts, floods, and heatwaves)".

2. lines 78-79, p.4: It would be good to specify the years covered by your review to contextualise the eight viral diseases declared as PHEICs by the WHO.

3. lines 87-95, p.4: Similar to the start of the Abstract and line 159 on p.7, a broader definition of vulnerable populations was initially given but later only gender differences were considered. This is also seen in the "Gender-sensitivity of data system studies" section starting on p.23. If this is the case, I would suggest making the focus on the gender-lens more explicit in the Abstract and Introduction, and adding a sentence or two to support this focus over other vulnerable populations.

4. Objectives and Research questions, p.7-8: It is unclear to me why there is a need to separate these two sections. The points are similar and can be combined under one section.

5. Table 1, p.9: Are the stages mentioned on p.7 in line with WHO Pandemic Preparedness Framework mentioned under "Pandemic Preparedness"?

6. Table 1, p.10: "AMR" is not defined.

7. lines 272-273, p.14: Two different spellings of "multi-disciplinary" were used.

8. lines 299-301, p.15: Is this definition of data systems the same as that mentioned in Table 1?

9. line 302-303, p.15: Can you clarify what you mean by "a pilot of data extraction in duplicate"?

10. Table 2, p.26-30: The "Aims" column seems unnecessary because all the details are captured in the other columns and all the papers are CEAs. I would suggest removing this column.

11. line 567, p.31: There is an extra space between "for" and "costing". There is also an extra space between "in" and "duplicated" on line 595 of p.33, and between "gender" and "-blind" on line 621 of p.37.

12. Table 3, p.32: The "Aims" column can be removed. For consistency with Table 2, you can add a "Intervention/comparators" column too.

6. PLOS authors have the option to publish the peer review history of their article (what does this mean? ). If published, this will include your full peer review and any attached files.

**Do you want your identity to be public for this peer review?** For information about this choice, including consent withdrawal, please see our Privacy Policy .

Reviewer #1: No

Reviewer #2: No

---

## [Decision Letter · Decision Letter 1]

31 Jan 2025

PGPH-D-24-02054R1

Gaps and Opportunities for Data Systems and Economics to Support Priority Setting for Climate-Sensitive Infectious Diseases in Sub-Saharan Africa: A Rapid Scoping Review

Dear Dr. Delight,

Thank you for submitting your manuscript to PLOS Global Public Health. After careful consideration, we feel that it has merit but does not fully meet PLOS Global Public Health’s publication criteria as it currently stands. Therefore, we invite you to submit a revised version of the manuscript that addresses the points raised during the review process.

We look forward to receiving your revised manuscript.

Kind regards,

Charin Modchang, Ph.D.

Academic Editor

Additional Editor Comments (if provided):

Thank you for addressing the reviewers' initial comments in your revised manuscript. However, one reviewer has raised additional concerns, particularly regarding the search terms used in your study. Please address the remaining comments accordingly.

Reviewers' comments:

Reviewer's Responses to Questions

**Comments to the Author**

1. If the authors have adequately addressed your comments raised in a previous round of review and you feel that this manuscript is now acceptable for publication, you may indicate that here to bypass the “Comments to the Author” section, enter your conflict of interest statement in the “Confidential to Editor” section, and submit your "Accept" recommendation.

Reviewer #1: (No Response)

Reviewer #2: All comments have been addressed

2. Does this manuscript meet PLOS Global Public Health’s publication criteria ? Is the manuscript technically sound, and do the data support the conclusions? The manuscript must describe methodologically and ethically rigorous research with conclusions that are appropriately drawn based on the data presented.

Reviewer #1: Yes

Reviewer #2: Yes

3. Has the statistical analysis been performed appropriately and rigorously?

Reviewer #1: Yes

Reviewer #2: N/A

4. Have the authors made all data underlying the findings in their manuscript fully available (please refer to the Data Availability Statement at the start of the manuscript PDF file)?

Reviewer #1: Yes

Reviewer #2: Yes

5. Is the manuscript presented in an intelligible fashion and written in standard English?

Reviewer #1: Yes

Reviewer #2: Yes

6. Review Comments to the Author

Reviewer #1: Thank you for your thoughtful response to my comments. A lot is improved in this version; several major comments are below. My primary concern now has to do with your search terms being perhaps too restrictive (or your objectives need to be re-clarified).

Major comments:

Positionality statement:

These are typically at the start of papers, and usually look more like this: https://pmc.ncbi.nlm.nih.gov/articles/PMC11007728/. I’m also concerned about the statement re: foregrounding the perspectives of African researchers when neither the first author nor senior author is African.

That being said, I don’t think every paper needs a positionality statement, nor is it required by this journal (to my knowledge). Furthermore, I appreciate the authors’ efforts to be responsive to this comment. Really, it’s the bibliometric analysis/temporal publication trends that to me is the trigger for needing a positionality statement.

I’d suggest removing the positionality statement and de-emphasizing the temporal publication trends part of your bibliometric analysis by focusing instead on the geographical publication trends, and including authorship trends as a sentence at the end of that paragraph (remove the temporal publication trends paragraph). Then when you discuss in the Discussion section, add a sentence acknowledging that neither the lead author nor senior author on this paper are African, however….(add the justification currently in the positionality statement).

If you do want to keep temporal publication trends in as a separate paragraph in the methods and results, then I think you need to re-vamp your positionality statement to be more in line with the one I shared.

Introduction

Line 89-90: This is absolutely true, but I’m not sure that it makes sense to link this to economic analyses. Addressing systematic oppression is critical to resolving health disparities and climate vulnerability, but it’s not necessarily cost-effective (i.e., justice is not necessarily advanced by linking it to economic arguments).

Line 158-164: Your objectives don’t mention the gender analysis, which leaves me still a bit confused about the point and scope of this paper. If the gender analysis is a major piece, then I would make that clearer in your intro (and possibly call it out as a sub-objective)

Results

Lines 460-467: Might be worth reflecting on the finding in lines 448-449 (inclusion of vulnerable groups in participatory surveillance)

Lines 474-477: I’m honestly quite shocked there were no economic evaluation studies on RVF or influenza. I’m thinking this comes down to including “climate change” and “pandemic preparedness” in the search terms—once diseases have been identified as a CSID with pandemic potential, is it necessary to require that climate change and/or pandemic preparedness appears in the text or in a MeSH term? I think you’re finding “false positive” evidence gaps because of this.

I think it’s important clarify whether your goal is to characterize the evidence among papers that “self-identify” as relevant to climate change and pandemic preparedness, or merely to characterize the available evidence. If the latter, I don’t think your search terms are well-suited. Effectively, I think it might have been better to have different search terms for the different aims of your study (bibliometric analysis, data systems, economic evaluations, priority settings, etc.)

Line 528: I think by virtue of your search terms, the priority setting papers would have to discuss either economic evaluations or data systems—that means you’ll have a biased sample and won’t see papers that use other means for priority setting.

Discussion

Line 613-614: Again, I think part of this is an artefact of your search terms: I don’t think you’ll find a lot on RFV if you require papers to refer to pandemic preparedness, but that doesn’t mean there’s gap in capacity or expertise

Minor comments:

Abstract

Line 42: should States be capitalized?

Methods

Table 1: are the inclusion criteria for health economic theme and data system theme restricted to the CSIDs listed in the row above?

Lines 209-211: I’m confused about what happened between the abstract review and the full text review. Were some of the abstracts excluded? Currently, it reads as if two people read abstracts, then read full texts, but I’m not sure why they would do that.

Lines 237-243: The bibliometric analysis needs to be mentioned in the introduction. It’s confusing to come across it for the first time in the Methods section.

Results

Lines 351-353: Suggest re-wording to: “Five CSIDs dominated the data system literature: Rift Valley fever (22 studies), Ebola virus disease (14 studies), influenza (12 studies), COVID-19 (12 studies), and Lassa fever (8 studies)” as it’s easier for the reader to parse

Line 389: capitalize One Health

Line 392: replace “extent” with “range,” as the second part of the sentence doesn’t tell me that data sharing is extensive, it just tells me the extent varies

Line 399: What about environmental surveillance?

Reviewer #2: The authors have addressed all my comments. Please note however, that there is a typo on line 86 on page 4: "declarations the WHO" should be "declarations by WHO".

7. PLOS authors have the option to publish the peer review history of their article (what does this mean? ). If published, this will include your full peer review and any attached files.

**Do you want your identity to be public for this peer review?** For information about this choice, including consent withdrawal, please see our Privacy Policy .

Reviewer #1: No

Reviewer #2: No

---

## [Editor Report · Decision Letter 2]

7 Apr 2025

Gaps and Opportunities for Data Systems and Economics to Support Priority Setting for Climate-Sensitive Infectious Diseases in Sub-Saharan Africa: A Rapid Scoping Review

PGPH-D-24-02054R2

Dear Ms Delight,

We are pleased to inform you that your manuscript 'Gaps and Opportunities for Data Systems and Economics to Support Priority Setting for Climate-Sensitive Infectious Diseases in Sub-Saharan Africa: A Rapid Scoping Review' has been provisionally accepted for publication in PLOS Global Public Health.

Best regards,

Charin Modchang, Ph.D.

Academic Editor

The authors have adequately addressed all reviewer comments, significantly improving the manuscript. The revised manuscript meets the journal's standards and is suitable for publication.